

# Seasonal climate signals preserved in biochemical varves: insights from novel high-resolution sediment scanning techniques

Paul D. Zander[1*], Maurycy Żarczyński[2], Wojciech Tylmann[2], Shauna-kay Rainford[3], Martin Grosjean[1]

[1]Institute of Geography & Oeschger Centre for Climate Change Research, University of Bern, Switzerland
[2]Faculty of Oceanography and Geography, University of Gdansk, Poland
[3]Institute of Plant Sciences & Oeschger Centre for Climate Change Research, University of Bern, Switzerland

*Correspondence to*: Paul D. Zander (paul.zander@giub.unibe.ch)

**Abstract.** Varved lake sediments are exceptional archives of paleoclimatic information due to their precise chronological control and annual resolution. However, quantitative paleoclimate reconstructions based on the biogeochemical composition

of biochemical varves are extremely rare mainly because the climate-proxy relationships are complex, and obtaining biogeochemical proxy data at very high (annual) resolution is difficult. Recent developments in high-resolution hyperspectral imaging (HSI) of sedimentary pigment biomarkers combined with micro X-ray fluorescence (µXRF) elemental mapping make it possible to measure the structure and composition of varves at unprecedented resolution. This provides opportunities to explore (seasonal) climate signals preserved in biochemical varves and, thus, assess the potential for annual resolution climate

reconstruction from biochemical varves.

   Here, we present a geochemical dataset including HSI-inferred sedimentary pigments and uXRF-inferred elements at very high spatial resolution (60 µm, i.e. > 100 data points per varve year) in varved sediments of Lake Żabińskie, Poland over the period 1966–2019 CE. We compare this data with local meteorological observations to explore and quantify how changing seasonal meteorological conditions influenced sediment composition and varve formation processes. Based on the dissimilarity

of within-varve multivariate geochemical time series, we classified varves into four types. Multivariate analysis of variance shows that these four varve types were formed in years with significantly different seasonal meteorological conditions. Generalized additive models (GAMs) were used to infer seasonal climate conditions based on sedimentary variables. Spring and summer (MAMJJA) temperature were predicted using Ti and total C ($R^2_{adj}$ = 0.55; cross-validated root mean square error (CV-RMSE) = 0.7 °C, 14.4 %). Windy days from March to December (mean daily wind speed > 7 m/s) were predicted using

mass accumulation rate (MAR) and Si ($R^2_{adj}$ = 0.48; CV-RMSE = 19.0 %). This study demonstrates that high-resolution scanning techniques are promising tools to improve our understanding of varve formation processes and climate-proxy relationships in biochemical varves. This knowledge is the basis for quantitative high-resolution paleoclimate reconstructions, and here we provide examples of calibration and validation of annual resolution seasonal weather inference from varve biogeochemical data.



## 1. Introduction

Quantitative paleoclimatic reconstructions are essential for understanding how the climate system functions (IPCC, 2013; Tierney et al., 2020). Spatially distributed, high-resolution paleoclimatic records are necessary to understand regional scale paleoclimate variability and to contextualize current climate change (Neukom et al., 2019; Consortium, 2013). Varved lake sediments have long been recognized as unique archives of climate (De Geer, 1908) because of their precise, annually-resolved, age control, and because of the wide variety of paleoenvironmental information preserved in varved sediments (Zolitschka et al., 2015). Numerous studies have identified close relationships between instrumental meteorological records and data obtained from varved sediments, in some cases at annual resolution, demonstrating the great potential of varves for high-resolution paleoclimatic reconstructions. The majority of these studies have related meteorological parameters with sedimentary variables in clastic varves that reflect transport of minerogenic material (Lapointe et al., 2020; Francus et al., 2002; Trachsel et al., 2010; Elbert et al., 2012). Interpretation of climate signals recorded in the sedimentary properties of biochemical and biogenic varves has proven much more challenging than in clastic varves due to more complex and often non-linear interactions between climate forcing, ecological and hydrochemical response, endogenic organic matter and mineral formation, and sedimentation (Zolitschka et al., 2015). An extensive compilation of more than 1000 varve-related publications (http://www.pages.unibe.ch/science/end-aff/varves-wg/varve-related-publications, last updated 4 April 2019) includes only three studies (Amann et al., 2014; Swierczynski et al., 2012; Tian et al., 2011) that report significant correlations between meteorological data and bulk geochemical data from biogenic varves. Accordingly, climate reconstructions from geochemical proxies of biogenic varves are extremely rare. More commonly, microfossil assemblages have been used to reconstruct climate variables from biogenic varves, however these techniques face challenges for sub-decadal scale analyses (Telford, 2019). Despite their widespread occurrence in temperate zones, biogenic and biochemical varves remain an under-utilized archive for high-resolution quantitative paleoclimate reconstructions.

During the past decades, high-resolution sediment scanning techniques, such as X-ray fluorescence (XRF) and reflectance spectroscopy have been increasingly used for environmental reconstructions, including quantitative sub-decadal climate reconstructions from varved (Amann et al., 2014; Trachsel et al., 2010; Lapointe et al., 2020) and non-varved lacustrine sediments (von Gunten et al., 2012; Boldt et al., 2015). The speed and resolution of these non-destructive scanning measurements (often 0.2-2 mm resolution) is impossible to achieve with conventional biogeochemical methods that require destructive sampling of sediment cores. In this study, we use cutting-edge micro X-ray fluorescence (µXRF) imaging (to measure elements) and hyperspectral imaging (HSI, sedimentary pigments) methods that improve upon commonly used linescan techniques by producing two-dimensional images of geochemical data at 60 µm resolution. This improvement is critically important for sub-annual and annual resolution analyses because annual layers can be delineated more precisely and consistently using images of geochemical data. Additionally, 60 µm resolution yields an average of ~100 data points per varve at our site, enabling detailed investigation of (sub)seasonal-scale geochemical variability.



Here, these high-resolution imaging spectroscopy techniques were applied to sediments of Lake Żabińskie to gain insights into climate-proxy relationships in biochemical varves and to investigate how recent climatic variability is recorded in varve composition. This site features varves with excellent preservation, high sedimentation rates (6 mm year$^{-1}$ during 1966-2019), and complex varve structures showing substantial year-to-year variations. We focus on a 54-year period (1966-2019) with exceptionally precise chronology, i.e. no uncertainty in the varve count chronology. These properties make the site ideal for studying the influence of weather on varve structure and composition. This study aims to answer the following research questions: 1) how are seasonal weather conditions recorded in biochemical varves, and 2) how can varve composition be used to reconstruct seasonal meteorological conditions? Results show that at Lake Żabińskie, sub-annual geochemical patterns are influenced by seasonal meteorological conditions, and varve composition can be used to infer temperatures in spring/summer and windiness during the ice-free season.

## 2. Materials and methods

### 2.1 Site description and core collection

Lake Żabińskie is a kettle-hole lake formed in the post-glacial landscape of the Masurian Lakeland in Poland (54.1318° N, 21.9836° E; Fig. 1). The basin is small (41.6 ha) and relatively deep (44.4 m), which promotes thermal stratification. Limnological data from 2011-2013 show that complete mixing of the water column occurs 0-2 times per year (Bonk et al., 2015). The catchment geology is mainly glacial till, sandy moraines and fluvioglacial sands and gravels. Anoxic and eutrophic conditions have led to good preservation of thick biochemical varves. Bonk et al. (2015) documented varve formation processes at Lake Żabińskie via limnological monitoring, sediment trapping, and microscopic and geochemical investigation of recent sediments. An annual cycle of sedimentation was described with algal blooms and Si deposition in spring, followed by calcite precipitation in spring and summer, then Fe and S enriched sediments in fall, and finally organic and lithogenic detritus (enriched in K and Ti) deposited in winter. Further investigations of the sedimentary record have documented changes to the lake mixing regime, trophy, and catchment erosion, with the most significant environmental changes occurring after major deforestation and the development of agriculture in the catchment during the 17$^{th}$ century (Zander et al., 2021; Żarczyński et al., 2019; Hernández-Almeida et al., 2017; Wacnik et al., 2016; Bonk et al., 2016).

Sediment cores used in this study were retrieved in 2012 (ZAB-12-1), and 2020 (ZAB-20-1) using an UWITEC gravity corer (ø90 mm). Resin-embedded sediment slabs and thin sections were produced following the method of Żarczyński et al. (2018). Core correlation was done on a varve-by-varve basis using images of the cores, resin blocks, and thin sections.

### 2.2 Chronology

Varve counting was performed on scanned images of thin sections and these counts were transferred to images of geochemical scanning data. Identification of varve boundaries was facilitated by previous studies on varve microfacies in Lake Żabińskie (Żarczyński et al., 2018), as well as μXRF elemental measurements. The beginning of the varve year was defined as the onset



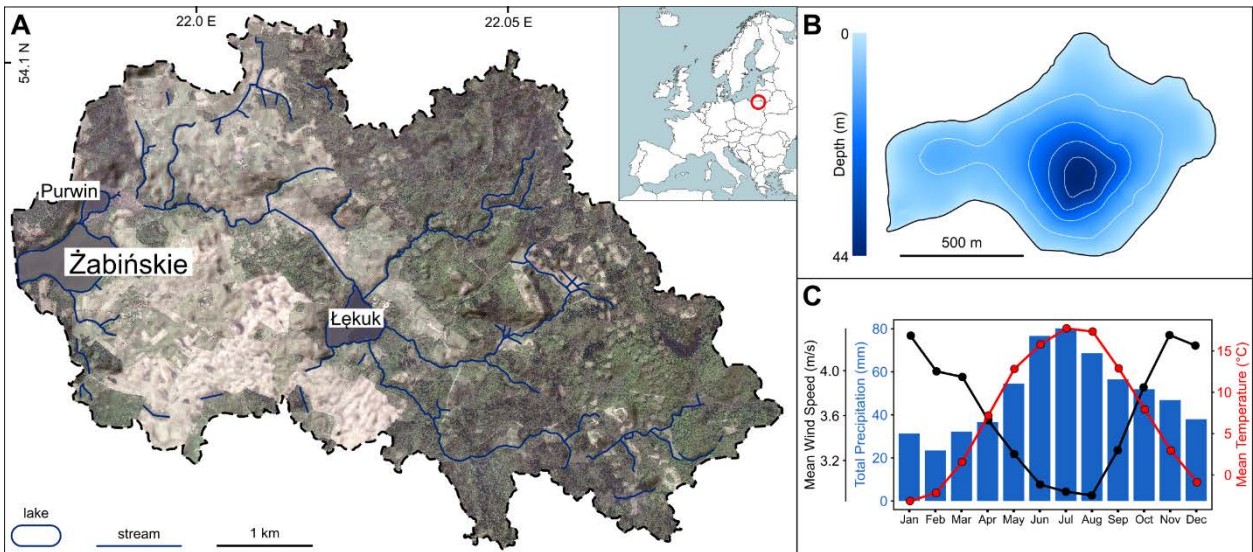

**Figure 1.** A) Orthophoto map of the Lake Żabińskie catchment (imagery from Polish Head Office of Geodesy and Cartography). Dark green colors represent forests; lighter colors represent cultivated areas. B) Bathymetric map of Lake Żabińskie. C) Summary of monthly meteorological data over the period 1966-2019

of calcite precipitation as this point can be consistently identified for every year in both scanning data and thin sections. Multiple researchers confirmed the varve count for the period 1966-2019 with no uncertainty. Varve thickness measurements were homogenized to account for differing sedimentation rates in the two cores by correcting the counting interval of each thin section to the actual depth covered by the same interval in the 2020 core. Dry bulk density was measured for each varve by sampling one cm$^3$ of wet sediment and measuring its weight after drying. Mass accumulation rates (MAR) were calculated by multiplying varve thickness and dry bulk density for each varve.

The varve count was tested using fallout from the 1986 Chernobyl incident as an independent time marker. Fallout was identified using $^{137}$Cs activities measured using gamma ray spectrometry (Tylmann et al., 2016). ZAB-12-1 was sampled for $^{137}$Cs activity in 3-year intervals. Individual varves were sampled from ZAB-20-1 for the years 2017, 2011, 1989-1984 and 1966 to better resolve the 1986 Chernobyl peak and to confirm baseline $^{137}$Cs activities during years not affected by Chernobyl fallout.

**2.3 Geochemical scanning measurements**

µXRF measurements were conducted on resin-embedded sediment slabs extracted from the 2012 and 2020 cores using a Bruker M4 Tornado. The scanner was equipped with a Rh X-ray source with voltage and current set to 50 kv and 300 µA, respectively. Counts were measured along the two-dimensional surface of the slabs with a measurement spot size of 20 µm and counting time of 20 ms/pixel. The measurement step (pixel size) was set to 60 µm.

Hyperspectral imaging was performed on the fresh ZAB-20-1 core using a Specim PFD-CL-65-V10E camera following methods described in Butz et al. (2015) and using the same acquisition settings and calculations as in Zander et al.





(2021). The scanning resolution (pixel size) was 60 μm. Relative absorption band depth (RABD) indices were used to quantify the abundance of sedimentary pigments. $RABD_{655–685max}$ represents total chloropigments (TChl), and $RABD_{845}$ represents bacteriopheopigments-*a* (Bphe). The RABD indices were calibrated to pigment concentrations ($\mu g/g_{d.s.}$; d.s. = dry sediments) using the calibration method described in Zander et al. (2021). TChl is used as a proxy for total algal productivity (Rein and Sirocko, 2002; Leavitt and Hodgson, 2002), whereas Bphe is produced by anoxygenic phototrophic purple sulfur bacteria and

is a specific biomarker for anoxic conditions overlapping with the photic zone (Butz et al., 2015; Sinninghe Damsté and Schouten, 2006). An additional spectral index, Rmean (mean reflectance/total brightness), was used as a proxy for calcite (Butz et al., 2017) to improve the alignment of HSI and μXRF data.

## 2.4 CNS elemental analysis

Total carbon (TC), total inorganic carbon (TIC), total nitrogen (TN), and total sulfur (TS) were quantified using a Vario El

Cube elemental analyzer (Elementar) equipped with a SoliTIC module and thermal conductivity detector following methods described in Żarczyński et al. (2019). Individual varves in ZAB-20-1 were sampled for these analyses. Total organic carbon (TOC) was calculated by subtracting TIC from TC. To account for post-depositional loss due to early diagenetic decay of organic matter, TOC and TN values were corrected using formulas developed by Gälman et al. (2008). The estimated TOC lost was added to TC values. All subsequent analyses were done with and without these corrections; the results are robust and

are weakly influenced by the correction.

## 2.5 Data analysis

Downcore profiles of μXRF and HSI data were established by averaging rows of pixels across 2 mm wide subsets of the rasterized geochemical data (thus each data point used for analysis represents a 60 x 2000 μm area). Data analysis was conducted in R 4.0.2 (R Core Team, 2020). The code and data used to generate plots and statistics reported in this paper is

available at http://dx.doi.org/10.48350/156383. The HSI pigment data (from the fresh ZAB-20-1 core) was aligned with the μXRF data (from resin-embedded slabs) using the location of varve boundaries manually identified on images of the core and resin-embedded slabs. This alignment was refined at the sub-varve scale using a dynamic time warping algorithm (Tormene et al., 2008) to maximize the correlation between Rmean (total reflectance; HSI) and Ca (μXRF) (Fig. S1). Settings were applied such that the data could not be shifted by more than one year (typically less).

To classify varve types (VTs), the package 'distantia' (Benito and Birks, 2020) was used to calculate the dissimilarity measure psi ($\psi$; Gordon and Birks, 1974), which measures the dissimilarity between pairs of multivariate time series (here the within-varve geochemical profiles). The ward.d2 hierarchical clustering algorithm was then used to identify groups of varves with a similar sequence of geochemical data through the year. Data were detrended, log-transformed and normalized prior to classification. To determine if the years defined by VTs experienced differing seasonal meteorological conditions, a

multivariate analysis of variance (MANOVA) test was performed using seasonal meteorological data from Kętrzyn, Poland, located 40 km west of the study site. Meteorological data were retrieved from the Polish Institute of Meteorology and Water



Management – National Research Institute using open data API and climate 0.9.1 R package (Czernecki et al., 2020). We considered a 15-month period from March to May the following year to account for the uncertainty of assigning the varve boundary (spring calcite precipitation) to a fixed point in the year. We focus on mean daily temperature, the 90th percentile of

daily precipitation, and the 90th percentile of daily mean wind speed based on the expectation that days with intense precipitation or wind would have a stronger effect on the sediments than mean seasonal values.

To further investigate relationships between seasonal meteorological conditions and varve composition, a redundancy analysis (RDA) was performed ('vegan' package, Oksanen et al., 2020). Meteorological variables were used as explanatory variables, and annual mean geochemical data from varve layers were response variables. Additionally, correlation coefficients

(Pearson's r) were calculated for monthly and seasonal meteorological variables and mean annual proxy data. The significance of correlations was assessed with p-values that were corrected for autocorrelation using the method of Bretherton et al. (1999) and were corrected for multiple testing using the false discovery rate approach of Benjamini and Hochberg (1995).

Generalized additive models (GAMs) were used to reconstruct meteorological parameters from sedimentary variables. The method is analogous to multiple linear regression, but GAMs have the advantage of utilizing flexible predictor

functions that can account for non-linear relationships between predictor and response variables. The target meteorological parameters for reconstruction were selected based on the results of the previous correlation analysis and analysis of variance, which identified temperatures in spring and summer, and windiness throughout multiple seasons as the most important variables driving variability in varve composition and structure. Predictor variables were selected based on the strength of their linear relations. Multiple combinations of possible predictor variables were evaluated and final models were selected based on

their predictive power, but also mechanistic process understanding and plausibility. Models were fit using the restricted maximum likelihood (REML) smoothness selection (Wood, 2011). Model skill was assessed by the 10-fold cross-validated root mean square error of prediction (CV-RMSE). Because this statistic can underestimate errors when data is autocorrelated (Telford, 2019), a split-period approach was also used to calculate RMSE, as well as additional calibration statistics reduction of error (RE), and coefficient of efficiency (CE) (Cook et al., 1994).

**3. Results and discussion**

**3.1 Chronology**

The varve count showed no age uncertainty to the depth of the 1966 CE varve year. Correlation of varves between multiple cores from different coring years provides further confidence in the varve count (Fig. S2). Average varve thickness in the period 1966-2019 was 6.0 mm. Fallout from the 1986 Chernobyl event, recorded in $^{137}$Cs activities, peaks in the varves of

1985 and 1986 (Fig. 2). The $^{137}$Cs activities in these two varves are indistinguishable within the measurement uncertainty (~8 Bq kg$^{-1}$). The similar activities in these two varves is explained by the fact that the accident occurred in late April 1986 (roughly coincident with the varve boundary). Additionally, some post-depositional diffusion of $^{137}$Cs is expected (Klaminder et al., 2012). These results provide independent validation of the accuracy of the varve count.





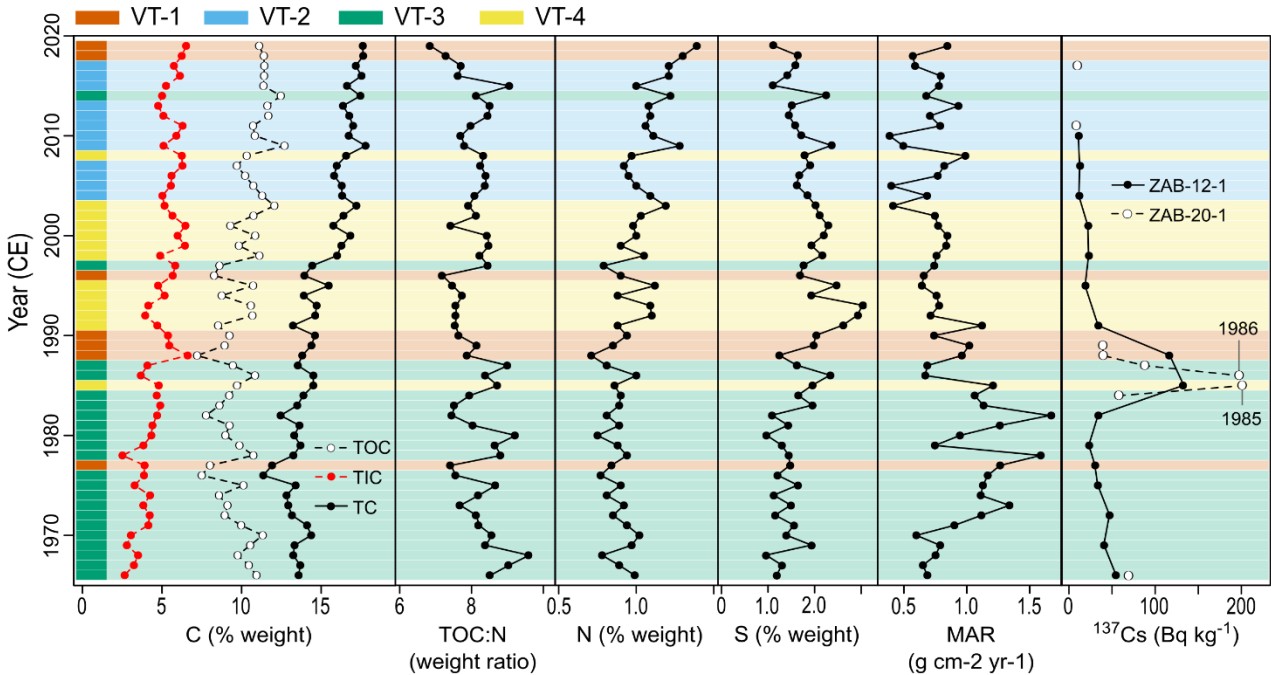

**Figure 2.** Results of CNS elemental analysis, mass accumulation rates, and 137Cs activities displaying the 1986 peak from the Chernobyl accident.

## 3.2 Geochemical results and interpretation

Varves in Lake Żabińskie are dominated by aquatic organic material and endogenic calcite. Several long-term trends (1966-2019) can be observed in the geochemical results (Fig. 2; Fig. 3). TOC, Ca and S show increasing trends toward the present while MAR, Fe, Si, and Mn decrease. The major increase in Ca is particularly notable, with much higher peaks after around 1992 indicating more calcite precipitation in the epilimnion. At the same time, Mn-rich layers become less frequent after 1992, most likely indicating less frequent seasonal mixing events. Mn layers are preserved during seasonal mixing of the water column that oxygenates the hypolimnion. Oxygenation of the bottom waters leads to precipitation of Mn that had previously been reductively dissolved in the normally anoxic hypolimnion (Scholtysik et al., 2020; Schaller and Wehrli, 1996). Less frequent mixing after 1992 is likely also responsible for increasing S towards the top of the core. TChl shows little long-term trend but significant year-to-year variability. Together, increasing TOC towards the present, stable TChl, and decreasing Si are interpreted as a result of a shift towards cyanobacteria and away from siliceous diatoms, as has previously been reported at Lake Żabińskie (Amann et al., 2014).

Aligning and averaging all varve years on a fractional varve-year scale enables visualization and description of a 'canonical' varve year that represents the typical sequence of geochemical variables through an annual cycle. (Fig. 4a). The 'canonical' varve year begins with calcite precipitation (as defined in our varve counting procedure), which occurs in the midst of peak primary production (recorded by TChl). Initial spring blooms of diatoms and other siliceous algae occur immediately

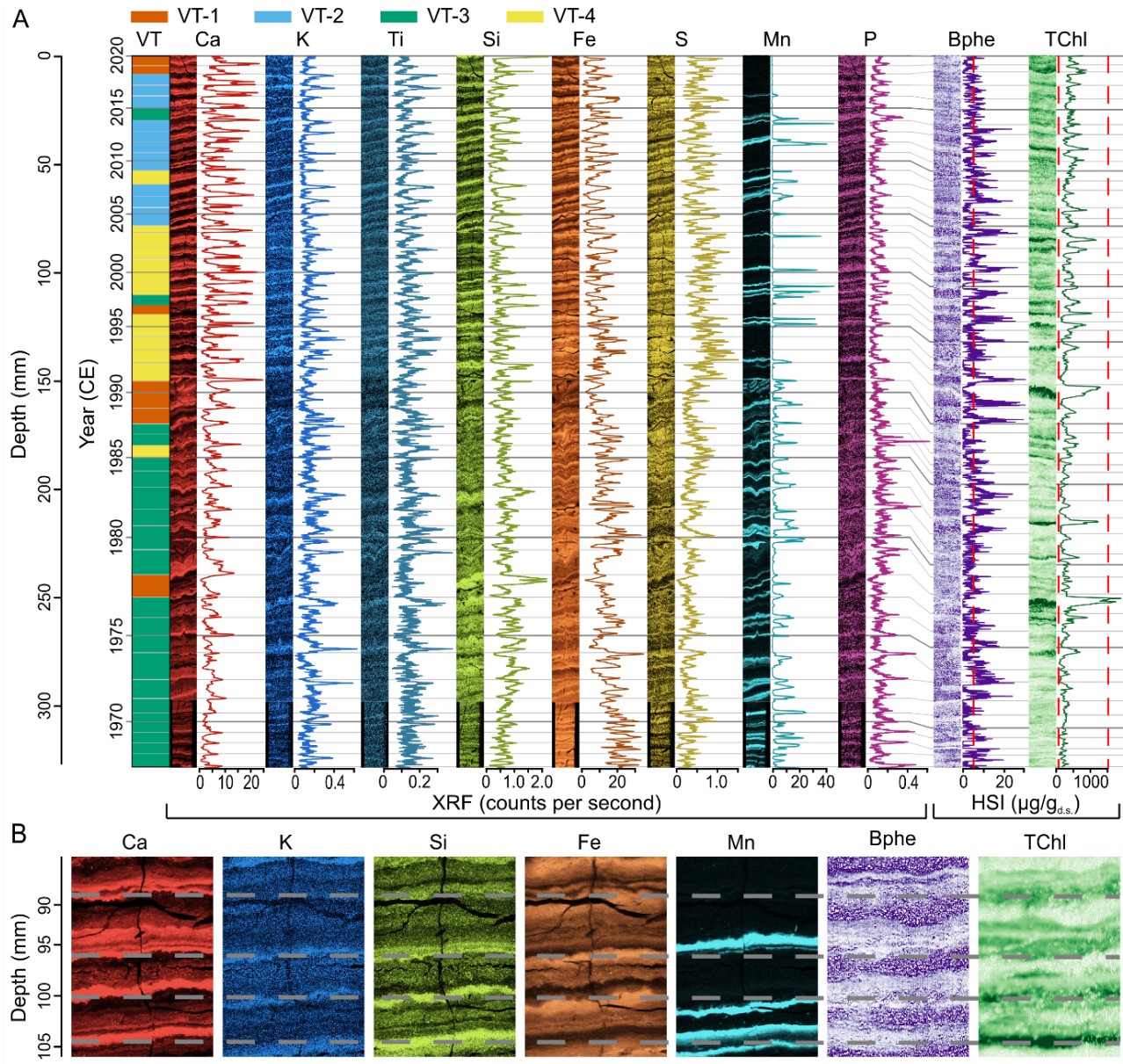

**Figure 3.** High-resolution geochemical data obtained from imaging spectroscopy (μXRF = micro X-Ray fluorescence, HSI = hyperspectral imaging). A) Composite sequence shown as both spatial maps and down-core profiles. Varve boundaries are shown as horizontal lines. Varve types (VTs) are plotted on the left side. μXRF data is from resin-embedded sediment slabs and the data from 1966-1990 is from the 2012 core, data from 1991-2019 is from the 2020 core. HSI data is from the wet 2020 core. Red dashed lines indicate the calibration range for pigment concentrations obtained using HSI. B) Close-ups of spatial distribution of elements and pigments in four varves.

after ice breakup, typically in March or April. Calcite precipitation follows after algal blooms uptake $CO_2$ and temperatures rise (Bonk et al., 2015). Based on this information, the varve year represents an annual cycle normally beginning in April or May. Calcite precipitation continues throughout the summer with one to three distinct calcite laminae preserved. P typically





tracks closely with Ca (Fig. 4a), suggesting co-precipitation of P with carbonates. TChl declines from high values at the start of the varve year and is low through late-summer and fall, before rising during winter deposition of fine organic detrital material and peaking during spring algal blooms marking the end of the varve year. Si counts follow a similar seasonal pattern as TChl, with major diatom blooms associated with the largest peaks in both Si and TChl. However, silicate minerals are also a major source of Si, and Si actually correlates better with K ($r = 0.36$, $p < 0.01$; Fig. S3) than TChl ($r = 0.24$, $p < 0.01$). Ti follows an opposite pattern compared to Ca with lowest values in the beginning of the varve year and maximal values near the end of the of varve year. High values of Ti denote the ice-cover period when mainly fine lithogenic detrital material is deposited. Bphe concentrations are generally low, leading to more measurement noise than the other proxies. Bphe shows a seasonal pattern with highest values occurring in late summer/fall, when the anoxic boundary remains shallow, before cooling in the epilimnion leads to mixing and lowering of the anoxic boundary (Bonk et al., 2015). This terminal stratification period likely features greater light penetration compared to more productive times of spring and summer, ideal for growth of anoxygenic phototrophic bacteria that require light penetration at the chemocline (Sinninghe Damsté and Schouten, 2006). Fe and S show a less-pronounced seasonal pattern than other elements, but tend to reach highest values in fall and winter. Fe often tracks Ti over the course of the year, indicating lithogenic detrital input is an important source of Fe. Mn has the most variable seasonal pattern with large peaks occurring at different points in the varve year; however, Mn peaks are generally absent in late summer/early fall, and some years show no Mn-rich layers (likely indicating a lack of deep mixing).

### 3.3 Classification of varve type

The results of the hierarchical clustering algorithm based on the dissimilarity measure of multivariate time series (Fig. S4 & S5) show distinct differences in the seasonal deposition of different elements and pigments (Fig. 4b), which correspond to different conditions in the lake and catchment.

Varve type 1 (VT-1; n = 7, Fig. 4b) occurs occasionally throughout the record and is characterized by low lithogenic input (Ti), and generally lower counts of redox sensitive elements Fe, S, and Mn, though Mn peaks are occasionally present. This pattern indicates that, in the years with VT-1, the water column was strongly stratified and calm, with minimal erosional input or sediment focusing bringing detrital lithogenic material to the lake center. Additionally, elevated values of TChl and high Si near the end of the varve year indicate high productivity at the onset of the following spring prior to calcite deposition.

VT-2 (n = 12, Fig. 4b) occurs exclusively since 2004 and is the most common varve type in the past two decades. This type is defined by the highest Ca values in the first half of the varve year, a distinct rise in Bphe, Fe, S and Mn in the second half of the varve year, and relatively high Ti values in winter. This pattern indicates years with a warm and productive epilimnion in summer, leading to extensive anoxia in fall, promoting growth of anoxygenic phototrophic bacteria and formation of iron-sulfides. Mn peaks in some years indicate mixing in spring or late-fall.

VT-3 (n = 22, Fig. 4b) dominates during the period 1966-1987, and is identified in only two years after 1987. This type is defined by generally high values of Fe, Mn and P, and low values of Ca and TChl. The pattern of Fe is variable. High





**Figure 4.** A) Average annual sequence of key geochemical variables across a varve year. Varve year begins in spring with calcite precipitation. B) Groups of annual time series determined from hierarchical clustering based on the dissimilarity measure ψ applied to the annual time series. Data plotted is means of varve types, with shaded regions representing 80 % of the data for each group.



values of Mn and Fe throughout the varve year suggest that complete mixing of the water column occurred more often during
these years. Detectable concentrations of Bphe in most years are evidence that strong thermal stratification and anoxic
conditions still occurred in summer/early-fall. Median values of TChl are low in VT-3 years, though there are some years with
large TChl peaks (algal blooms). Oxygenation of the water column may have limited TChl preservation (Leavitt and Hodgson,
2002).

VT-4 (n = 13, Fig. 4b) occurs primarily from 1991-2003 and is characterized by high Ca, TChl and S values. Mn is
250 variable with very high peaks occurring in some years and no Mn in other years. Mn peaks tend to occur either in summer or
late fall/early winter. Fe and Ti peak in winter. Bphe is again present mainly in fall. This varve type has higher S counts than
the other types, likely indicating strongly reducing conditions in the hypolimnion and sediments (Håkanson and Jansson, 1983).

### 3.4 Relationships between varve composition and meteorological conditions

A MANOVA test was applied to test the hypothesis that the four different varve types (VTs) were formed in years with
255 different seasonal meteorological conditions. The MANOVA test yields a significant result (p = 0.001), allowing us to reject
the null hypothesis. Specifically, temperatures in spring (MAM), summer (JJA) and fall (SON) and windiness in spring and
fall were significantly different (p < 0.05) in the years corresponding to the four varve types (Fig. 5). The differences in the
meteorological data are consistent with the geochemical patterns defined by the VTs. VT-2 shows a strong effect of weather
conditions with consistently warmer temperatures and less wind in these years. The geochemical character of VT-2 indicates
productive, well-stratified and anoxic conditions with intensive calcite precipitation (Fig. 4b). This suggests calcite
precipitation is strongly influenced by epilimnetic temperatures, consistent with research in other lakes (e.g. Stabel, 1986).
VT-3 is associated with cooler temperatures and more wind (Fig. 5), which promoted more frequent lake mixing and lowered
lake productivity, as evidenced by high Mn, P and Fe preservation and low TChl in these varves (Fig. 4b). Varve types 1 and
4 show greater variability in the meteorological data (Fig. 5), with no clearly interpretable effect of weather on their formation.

Relationships between meteorological conditions and varve composition were further explored using a redundancy
analysis (RDA; Fig. S6) and correlation analysis between annual mean values of sedimentary variables and monthly/seasonal
meteorological variables (Fig. 6; Table 1). The results of the RDA indicate that 46.8 % of the variance in the response variables
(sedimentary variables) is shared with the explanatory variables (seasonal meteorology). The first RDA axis explains 28 % of
the variance and shows strong (opposing) effects of temperature and wind on several sedimentary variables, particularly MAR,
Ca and TC. Precipitation has very little shared variability with the sedimentary data.

The monthly correlation analysis (Fig. 6) provides insight into how relationships between sedimentary variables and
weather vary through the varve year. Generally, temperatures are positively correlated to Ca, TC, TIC and TN, and negatively
correlated with lithogenic elements (Ti, K). Spring and summer temperatures show the strongest relationship with the
composition of varve layers; in particular TC is well correlated with MAMJJA temperatures (r = 0.69, $p_{adj}$ = 0.004; Table 1).
These correlations most probably reflect a combination of temperature-related mechanisms whereby warmer temperatures
increase the duration of the growing season (shortened ice cover), increase algal growth rates (Butterwick et al., 2005), and

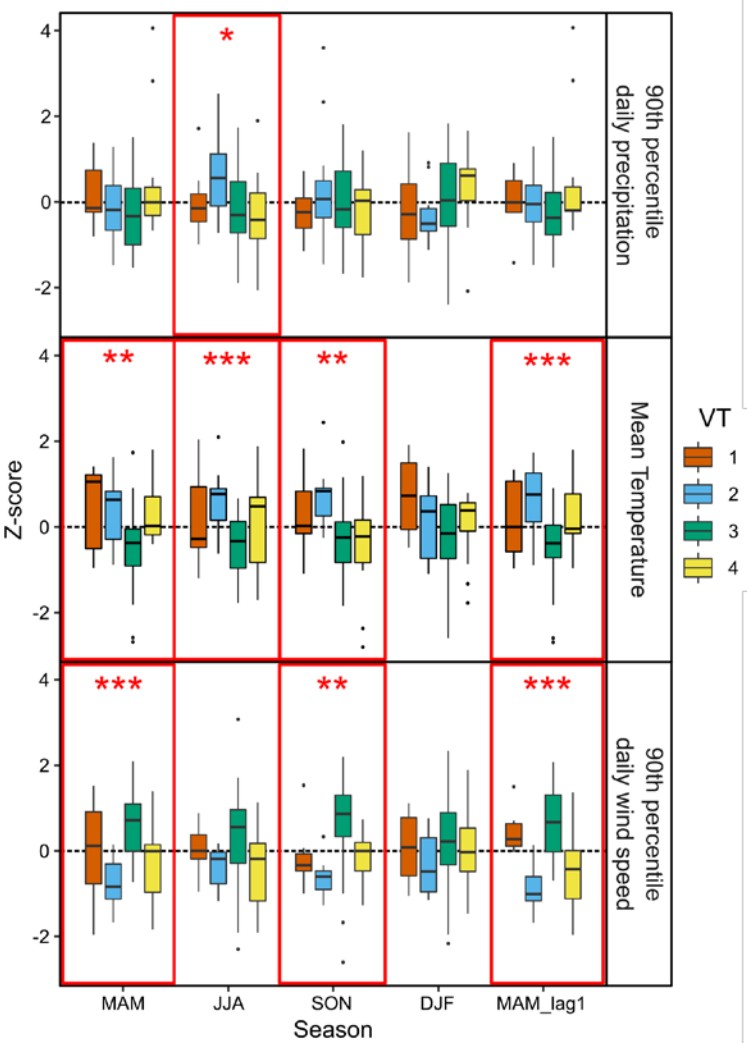

**Figure 5.** Boxplots of seasonal meteorological variables separate by varve types (VT). Red boxes and * symbols indicate meteorological variables with significantly ($p < 0.05$) different means for the four VTs, based on an analysis of variance test (* $p < 0.1$, ** $p < 0.05$, *** $p$ 280 $< 0.01$).

lower the solubility of carbonates in the epilimnion (Plummer and Busenberg, 1982). Calcium carbonate solubility is controlled directly by water temperature and secondarily through $CO_2$ uptake from algal production (Stabel, 1986). Additionally, stronger thermal stratification and more extensive and persistent anoxia can increase carbon burial (Bartosiewicz et al., 2019). Negative correlations with Ti and K are likely driven by a dilution effect, whereby warmer epilimnetic 285 temperatures and longer growing seasons lead to increased production of endogenous carbonate and organic matter. This increase in endogenous material results in lower concentrations of lithogenic components (Ti) when considering the mean composition of a varve. The geochemical patterns of VT-2 and VT -3 illustrate this mechanism (Fig. 4b). In VT-3, which formed during cooler years, high Ti values make up a greater portion of the varve year compared to VT-2 (warmer years).





**Figure 6.** Correlation matrices of Pearson correlation coefficient (r) for sedimentary variables (annual mean values) and A) $90^{th}$ percentile of daily precipitation for each month, B) mean monthly temperature, and C) $90^{th}$ percentile of daily mean wind speeds for each month. Significance of correlations is identified by * symbols (* $p_{adj} < 0.1$, ** $p_{adj} < 0.05$, *** $p_{adj} < 0.01$).

Therefore, annual mean values of Ti tend to be lower in warmer years. Our results do not show a significant correlation between TChl and spring temperature, as was previously found by Amann et al. (2014) at Lake Żabińskie over the period 1907-2008 CE (r = 0.36, $p_{adj} < 0.05$ for annual resolution data). In our dataset the correlation is positive but not statistically significant (r = 0.20, $p_{adj} = 0.30$; Table 1), an example of how climate-proxy relationships are not always stable in time (Blass et al., 2007).





**Table 1.** Correlation matrix of Pearson correlation coefficients for selected meteorological variables and data from varves (annual mean values of geochemical data). Bold values indicate significant correlations ($p_{adj} < 0.05$). P-values were corrected for autocorrelation (Bretherton et al., 1999) and the false discovery rate (Benjamini and Hochberg, 1995).

| | Temperature | | | | | | 90th percentile Wind | | | | |
|---|---|---|---|---|---|---|---|---|---|---|---|
| | MAM | JJA | SON | DJF | Ann | MAMJJA | MAM | JJA | SON | DJF | Wind Days Mar-Dec |
| Ca | **0.58** | **0.45** | 0.31 | 0.29 | **0.58** | **0.61** | -0.32 | -0.24 | -0.36 | -0.21 | -0.50 |
| Fe | -0.41 | -0.33 | -0.28 | -0.39 | **-0.55** | -0.44 | 0.31 | 0.12 | 0.30 | 0.02 | 0.41 |
| Mn | -0.08 | -0.17 | -0.15 | -0.26 | -0.28 | -0.14 | **0.46** | 0.09 | 0.00 | -0.02 | 0.26 |
| Si | **-0.44** | **-0.46** | -0.30 | -0.01 | -0.36 | **-0.53** | **0.48** | **0.45** | 0.49 | 0.40 | **0.62** |
| P | -0.20 | -0.31 | -0.10 | -0.30 | -0.37 | -0.29 | **0.45** | 0.25 | 0.38 | 0.14 | 0.41 |
| S | 0.26 | 0.29 | -0.07 | 0.09 | 0.20 | 0.32 | -0.31 | -0.16 | -0.14 | -0.11 | -0.33 |
| K | **-0.57** | -0.35 | -0.32 | 0.11 | -0.30 | **-0.55** | 0.28 | 0.36 | **0.50** | 0.41 | **0.45** |
| Ti | **-0.66** | -0.34 | **-0.39** | -0.04 | **-0.44** | **-0.60** | 0.24 | 0.27 | 0.36 | 0.25 | 0.36 |
| Bphe | 0.03 | 0.12 | 0.13 | -0.11 | 0.01 | 0.09 | 0.06 | -0.30 | -0.08 | 0.07 | -0.05 |
| TChl | 0.20 | 0.28 | 0.25 | 0.20 | 0.34 | 0.28 | -0.31 | 0.00 | -0.22 | -0.07 | -0.24 |
| TOC | **0.40** | 0.35 | 0.29 | -0.18 | 0.19 | **0.44** | -0.41 | -0.33 | -0.36 | **-0.44** | **-0.55** |
| TIC | **0.46** | **0.51** | 0.27 | **0.47** | **0.66** | **0.56** | -0.42 | -0.29 | -0.38 | -0.14 | **-0.44** |
| TN | **0.52** | **0.46** | 0.27 | -0.05 | 0.34 | **0.58** | -0.40 | -0.27 | -0.27 | -0.33 | **-0.50** |
| TC | **0.59** | **0.59** | 0.39 | 0.17 | **0.57** | **0.69** | **-0.58** | -0.44 | -0.51 | -0.42 | **-0.70** |
| MAR | -0.18 | -0.40 | -0.14 | 0.06 | -0.17 | -0.33 | **0.51** | 0.37 | **0.52** | 0.56 | 0.63 |

Correlations between windiness and sedimentary variables show mostly an opposing pattern to temperature. The variable with the most consistent strong relationship to windiness is MAR (mass accumulation rate). These positive correlations suggest a strong effect of sediment focusing due to wind driven turbulence in shallow parts of the lake, as has been observed in other lakes with varved sediments (e.g. Nuhfer et al., 1993; Roeser et al., 2021). The importance of sediment focusing at Lake Żabińskie was previously identified by higher than expected [210]Pb and [137]Cs inventories (Tylmann et al., 2016). Si also shows significant correlations with windiness. Increased Si during windy years can be attributed to a combination of three mechanisms: 1) resuspension of lithogenic material (silicate minerals), 2) resuspension of siliceous algae remains (Raubitschek et al., 1999), 3) increased production of siliceous algae due to increased availability of nutrients such as N, P and Si in the epilimnion due to mixing of the water column (Conley et al., 1993). Wind-driven mixing also strongly affects redox conditions within the lake. Positive correlations between wind and Mn, Fe, and P can be attributed to deep mixing events driven by wind, particularly in spring (Naeher et al., 2013). Lowering of the anoxic boundary results in (re)precipitation of Mn and Fe (hydr)oxides and improves preservation of these elements in the sediments through 'geochemical focusing' (Scholtysik et al., 2020; Schaller and Wehrli, 1996). This geochemical focusing is also an important contribution to higher MARs during windier years (i.e. VT-3; Fig. 4b). The importance of wind shear on Lake Żabińskie biogeochemical cycling has previously been identified in the studies of the Holocene sedimentary record of Lake Żabińskie (Zander et al., 2021; Żarczyński et al.,



2019; Hernández-Almeida et al., 2014). A significant negative correlation between bacteriopheophytin and July windiness could be explained by increased turbidity during windy summers (and limited light at the chemocline) and/or a lowering of the anoxic boundary below the photic zone; both effects would limit growth of anoxygenic phototrophic bacteria. Notably weaker

correlations between sedimentary variables and February wind are attributed to the fact that ice cover is consistently extensive throughout this month.

The correlation analysis (Fig. 6, Table 1) highlights proxies that are potentially suitable for paleoclimate reconstructions and for which seasons they are most sensitive. Accordingly, warm season temperatures can be reconstructed using a combination of Ti and TC (or other carbonate/organic proxies), while wind can be reconstructed using MAR, Si, K or

P though there is not a strong seasonal signal in the correlations. Bphe shows potential for reconstructing winds specifically in summer, however the Bphe data is noisy due to low concentrations, making this signal weak and less consistently reproducible compared to the other proxies (Fig. S7). Over the Holocene, Bphe concentrations were closely linked to forest cover surrounding the lake, which affects wind exposure (Zander et al., 2021).

### 3.5 Temperature and wind reconstructions using generalized additive models

Based on the correlation analysis and investigation of relationships between varve types and meteorological variables, we selected spring and summer (MAMJJA) temperature and the number of windy days (mean daily wind speed > 7 m s$^{-1}$) from March to December as the most suitable targets for reconstruction. We chose the March to December timeframe for the wind model because this is the typical ice-free period, and therefore is expected to be most susceptible to wind effects. The threshold for a wind day was selected based on the 95th percentile of daily wind speeds within a year to emphasize the importance of

extreme wind events. We use the 95th percentile here rather than the 90th percentile used for monthly and seasonal analysis because the March to December period is longer, making higher percentiles less susceptible to outliers.

Generalized additive models (GAM) were used to predict meteorological conditions based on varve data at annual resolution. We found TC and Ti provided significant predictive power for MAMJJA temperature and a GAM fit to these variables explains 56.4 % of the deviance in MAMJJA temperatures ($R^2_{adj}$ = 0.55; Fig. 7). Both predictor variables are

significant (p < 0.05). Partial effect plots (Fig. S8) show the relationships between temperature and predictor variables are nearly linear. Residual analysis confirms the good fit of the model with no trends or significant autocorrelation apparent. The model captures the trends in temperature well, however it appears to underestimate extreme variations (particularly cold or warm years). The 10-fold cross-validated root mean square error (CV-RMSE) is 0.69 °C (14.4 % of the range), and RMSE calculated based on a split-period approach is similar (0.68-0.80 °C; Table S1). Consistently positive values of RE (reduction

of error) and CE (coefficient of efficiency) indicate that the calibration model has predictive skill (Cook et al., 1994; Table S1). These calibration statistics provide evidence that this calibration may be viable for paleoclimate reconstruction.

Wind days were successfully reconstructed using a model with predictors MAR and Si. The years 1993 and 1994 were removed prior to fitting the model because wind data were missing from these years. This model explains 49.6 % of the deviance in wind days ($R^2_{adj}$ = 0.48; Fig. 7) and both MAR and Si are significant predictors (p < 0.05). Partial effect plots show





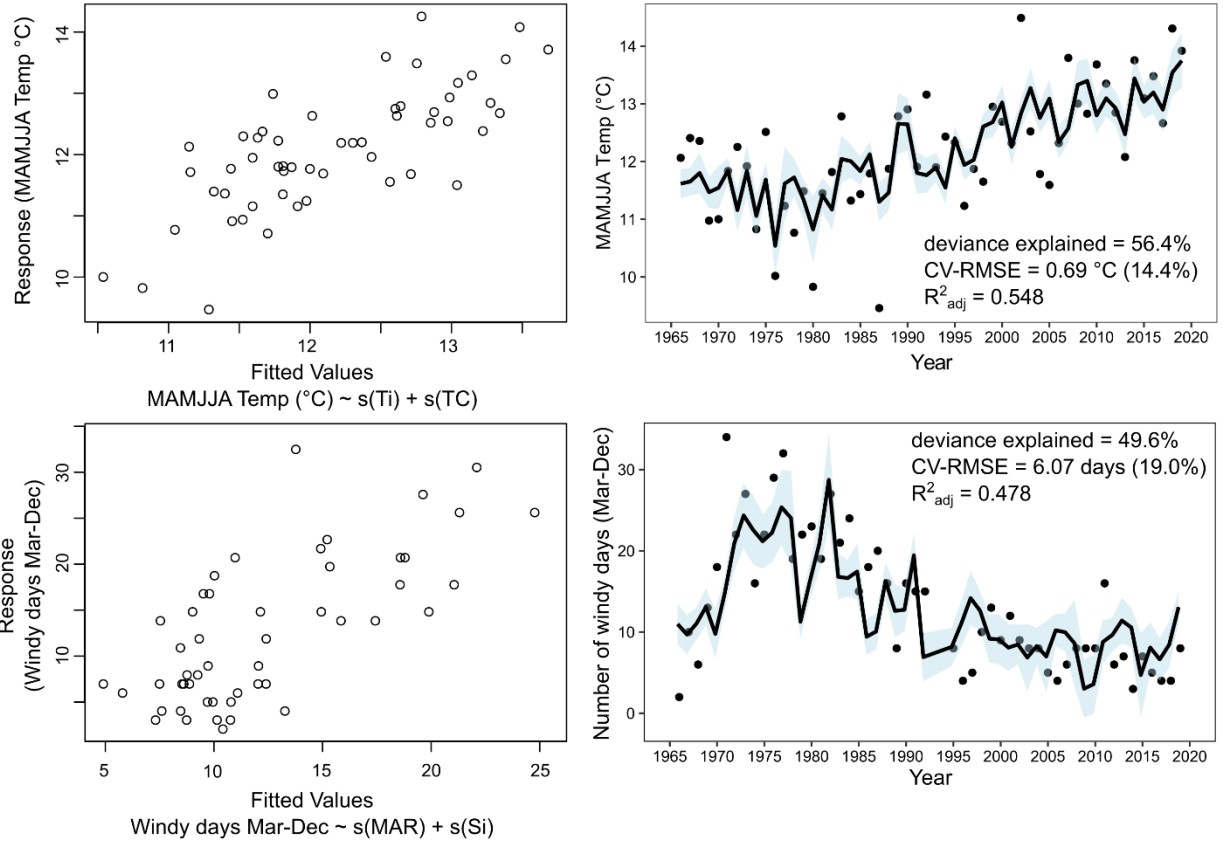


**Figure 7.** Results of generalized additive models used to predict spring and summer temperature and number of windy days from March to December (daily mean wind speeds > 7 m/s).

that the relationship between the predictor variables and wind days are approximately linear (Fig. S9). Residual analysis shows no strong trends, but the distribution appears slightly non-normal, suggesting the model fit may be flawed. Wind speeds at

Kętrzyn station show markedly higher values from 1970-1985, and the model is able to capture this trend well, however very low wind speeds in 1966 and 1968 are not well captured by the model. The model has a 10-fold CV-RMSE of 6.1 days (19.0 %), though this error increases substantially when using a split-period method (22.9 - 48.0 %, Table S1). The RE ranges from -0.28 to 0.61, depending on the selection of the calibration period (Table S1), and the CE is consistently negative. These statistics indicate that despite the good fit over the full calibration period, the model has relatively weak predictive skill. The

short study period, and major shifts in mean values between calibration and verification periods, could lead to unrealistically pessimistic results from the split-period validation. Additionally, wind speeds are more spatially variable than temperature, thus it is possible that local variations between wind speeds at Lake Żabińskie and the Kętrzyn station increase the model error. Further research over longer periods is needed before this model can be applied for paleoclimate reconstruction.



## 4. Discussion

### 4.1 Challenges for high-resolution climate reconstruction

Quantitative climate reconstructions from geochemical proxies of biogenic varves are extremely scarce in the published literature (as shown by the PAGES Varve Working Group compilation: http://www.pages.unibe.ch/science/end-aff/varves-wg/varve-related-publications). Our results show that novel scanning techniques with much higher spatial (and temporal) resolution offer new possibilities. However, several challenges concerning high-resolution climate reconstructions from

biogenic varves remain and are worthy of discussion. Measurements of sedimentary variables include some degree of noise that generally becomes more significant as resolution is increased (Blass et al. 2007). However, we find good reproducibility of our data across different cores, suggesting that a 60 x 2000 μm measurement area is large enough to produce a robust signal (Fig. S7). Uncertainty in assigning temporal values to data within a single varve remains a limitation because deposition rates are highly variable throughout the year. Determining the season of deposition of certain lamina is possible (i.e. the first calcite

lamina is deposited in April-June), but temporal uncertainty of a couple months or even more should be expected. Another important challenge is covariance of target meteorological variables. The climate variables used for reconstructions in this study (MAMJJA temperature and Mar-Dec wind days) are correlated (r = -0.52, $p_{adj} < 0.05$; Fig. S10), and both are significantly correlated (with opposing signs) with multiple sedimentary variables (Table 1). This covariance confounds interpretation of the proxy data because changes in a single variable (for instance, TC) may be influenced by both temperature and wind.

Autocorrelation of both proxy and meteorological time series also presents challenges when evaluating the significance of correlations and model performance (Telford, 2019). We account for autocorrelation by using the adjusted-n method for significance tests (Bretherton et al., 1999) and by using a split-period approach to calculate RMSE in addition to the cross-validated RMSE (Table S1). A longer calibration period would strengthen confidence in the calibration model; however, age uncertainty would require smoothing or aggregating data to lower temporal resolution, preventing annual resolution analysis.

The short 54-year calibration period increases the possibility that climate-proxy relationships identified here are not stable over longer time periods (Blass et al., 2007). This is especially problematic for sites that have experienced major environmental changes (often human-induced) outside the calibration period, as is known for Lake Żabińskie (Hernández-Almeida et al., 2017). Future work could investigate the climate-proxy relationships studied here over longer time-scales.

Non-climatic factors, such as human activity in the catchment, diagenetic effects, internal variability in

biogeochemical cycling, etc., also influence varve structure and composition at a variety of timescales. The proposed temperature reconstruction in this study assumes variations in TC are driven largely by temperature (via the effects of temperature on algal production and calcite precipitation). However, nutrient levels also likely influence the carbon contents of the sediments (Fiskal et al., 2019). Diatom-based P reconstructions show a slight increasing trend over the period 1966-2010; however, phosphate fertilizer use in the region peaked in the 1970s and has declined since then (Witak et al., 2017).

Sediment P also shows a decreasing trend over the study period (Fig. 3) and a negative correlation with TC (Fig. S11). Additionally, agricultural activities around Lake Żabińskie have decreased since 1950, leading to afforestation of abandoned





fields and reduced soil erosion in the catchment (Wacnik et al., 2016). Based on these trends, it appears more likely that increasing TC over the study period is related to warming temperatures rather that nutrient input to the lake. Recent reductions in soil erosion and afforestation around the lake (which reduces wind exposure) likely also influence MAR, which could be

one reason why the wind reconstruction does not perform well based on the split-period validation.

## 4.2 Potential for high-resolution climate reconstruction

Despite the aforementioned caveats, our results suggest that geochemical variables, particularly from high-resolution scanning techniques, are a promising tool for high-resolution quantitative climate reconstructions from biochemical varves. The high-resolution data obtained from imaging spectroscopy, thick varves at Lake Żabińskie, and annual certainty of the varve count

provide a unique opportunity to test relationships between seasonal meteorological conditions and varve composition. We present an innovative method to define varve types objectively using a multivariate time series clustering approach based on sub-annual variations in geochemistry. This technique may be considered a quantitative and time-efficient alternative (or complement) to microfacies approaches to varve classification (e.g. Żarczyński et al., 2019). We demonstrate that varves with different sub-annual geochemical time series formed during years with significant differences in meteorological conditions

(Fig. 5). The results of this analysis inform our interpretations of how meteorological variability is recorded in biochemical varves of Lake Żabińskie.

High-resolution imaging spectroscopy techniques offer several important advantages for sub-annual and annual resolution analyses of varves. The 60 μm resolution enabled us to study the annual cycle of sedimentation in great detail and observe significant differences in the year-to-year sequence of geochemical parameters. Images of geochemical data provide

information about the spatial nature of geochemical variations that is not available from linescan data (i.e. distinguishing nodules from continuous layers). XRF core-scanners such as ITRAX use an XRF beam that is 20 mm wide (Rothwell and Croudace, 2015), and unless laminations are perfectly parallel to this beam, some degree of smoothing (mixing) often occurs due to this wide beam area and the often undulated nature of varve boundaries. We are able to minimize this problem by using 2 mm wide subsets of the imaging data. Generally good reproducibility of the data is demonstrated by measurements done on

correlated sections of cores taken 8 years apart (Fig. S7). Additionally, the high-resolution images make it possible to delineate varve boundaries consistently and precisely. This is also important for annual resolution analyses – small offsets in the location of varve boundaries can lead to significant errors in mean annual values calculated from varve layers. In this study, due to the multi-mm-thick varves it was possible to sample annual layers for destructive analyses; however, at sites with thinner varves non-destructive scanning techniques are essential for annual resolution analyses.

The spring and summer temperature model shows strong potential for paleoclimate reconstruction at this site, and possible other sites with similar calcareous varves. The CV-RMSE of 0.69 °C is greater than other published reconstructions based on biogeochemical proxies from lake sediments (Amann et al., 2014; von Gunten et al., 2012), but this is because our calibration is at annual resolution, whereas the cited studies used smoothed sub-decadal data. Our RMSE is lower than those typically reported from climate transfer functions based on microfossil assemblages (Heiri et al., 2003), which are the typical





approach for quantitative climate reconstruction from biogenic varves. In contrast to microfossil approaches that require large samples and time-consuming analyses, geochemical data are quickly measured at high-resolution using non-destructive techniques such as XRF and reflectance spectroscopy. This makes it possible to efficiently apply these techniques to more sites, over longer time periods, and at higher resolution. We propose that these spectroscopic techniques have untapped potential for quantitative climate reconstructions from biochemical varves, particularly at high (annual) resolution. The calibration model presented here should be tested over longer time-scales, but our results provide the foundation for quantitative climate reconstruction from biochemical varves at Lake Żabińskie, and other sites with similar varve-formation processes.

## 5. Conclusions

The results of this study demonstrate the potential of high-resolution spectroscopy imaging techniques to enhance our understanding of sub-varve scale sedimentary processes and the relation to seasonal climate variables. The sequence of geochemical variables through the course of the varve year was shown to be influenced by changing seasonal meteorological conditions. Correlation analysis identified spring and summer (MAMJJA) temperatures, and windiness during the ice-free period as meteorological variables with the greatest potential for proxy-climate calibration from these sediments. Generalized additive models were applied to reconstruct these two variables at annual resolution over the study period (1966-2019). Total C and Ti were used as predictors for MAMJJA temperature and yielded a reconstruction with good prediction performance (CV-RMSE = 0.69 °C, 14.4 %). Mass accumulation rate (MAR) and Si were used to predict March-December wind days (CV-RMSE = 6.1 wind days, 19.0 %). Split-period validation provides evidence that the temperature reconstruction model has predictive skill and could be applied outside the calibration period, whereas the wind reconstruction model warrants caution if applied beyond the calibration period. Our results provide a rare example of quantitative climate calibration based on bulk geochemical data from biochemical varves with implications for paleoclimate reconstructions at other sites with calcareous varves. The approach used in this study can be applied to other sites with varved sediments to generate high-resolution paleoclimate reconstructions. Ultra-high-resolution spectroscopy imaging techniques, such as those applied in this study, show great potential for a variety of paleoenvironmental reconstructions including archives beyond lake sediments such as speleothems, fossils, tree ring cores and many others.

*Author contributions.* PDZ: Conceptualization, Formal analysis, Investigation, Writing - original draft, Visualization. MZ: Conceptualization, Investigation, Writing - Review & Editing, Resources, Visualization. WT: Writing - Review & Editing, Resources, Supervision, Funding acquisition, Project administration. SR: Writing - Review & Editing, Investigation. MG: Writing - Review & Editing, Resources, Supervision, Funding acquisition, Project administration.



*Code and data availability*. Data and code required to reproduce plots and statistics reported in this manuscript are available at the Bern Open Repository and Information System (BORIS; http://dx.doi.org/10.48350/156383).

*Competing interests.* The authors declare no competing interests


*Acknowledgements*. This work was funded by Swiss National Science Foundation grant 200021_172586 and Polish National Science Centre project NCN 2015/18/E/ST10/00325. Joanna Piłczyńska assisted with lab analyses.

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
