# Peer review of "Seasonal climate signals preserved in biochemical varves: insights from novel high-resolution sediment scanning techniques"

_Climate of the Past, 2021_

## Author Response (AR1)

**Authors' Response to RC1**

**General comments**

The manuscript by Zander et al. presents high-resolution geochemical records from biochemical varve sequence of Lake Zabinskie. The topic is very interesting and timely. The use of biogenic content as climatic proxy is underrepresented among varve studies, nearly missing, due to their more complex nature compared to for example clastic varves or minerogenic content of mixed varves. Exploring biochemical content of varved sediments provide new insights to climate studies and new opportunities for high-resolution climate reconstructions, not only for new locations but also for different seasons. Zander et al. use state-of-the-art methods and present high quality data with comprehensive statistical analyses. The manuscript is well written and the figures and tables are of high quality. The interpretations are logical and justified by data. I have only few minor comments to improve the manuscript and one technical issue related to manuscript structure.

Thank you for your comments and review of our manuscript. We feel the comments can be addressed in a revised version of the manuscript. Our responses are provided in red text.

**Specific comments**

Line 98 How many researchers calculated varves

Three researchers identified the varves, this has been added to the text. (Line 96)

Line 190 How about cultural S? If excluded in Lake Zabinskie, shortly comment. In addition, it seems from supplementary Figure 2 that varve characteristics change at 1965 where the analyses of this paper begins. It is out of the topic to discuss older sediments, but could you very briefly mention the reason for the change (anthropogenic?), so that reader would understand where the record starts, from what conditions related to anthropogenic activities. If known. With this background information, reader would have a better perspective on local conditions and hence easier to evaluate the data related to the elements sensitive to anthropogenic activities.

We interpret sulfur to be primarily of natural origin at this site, though to some degree it may be affected by human activities. There are no nearby industrial sources of S, though long-distance atmospheric transport from anthropogenic sources, soil erosion, and fertilizers could contribute some amount of sulfur to the lake. However, we think the deposition and preservation of S in the sediments is mainly controlled by redox and microbial processes.

Your observation is correct that there is a slight shift in varve characteristics around the time our study period begins. Generally, the varves in the years preceding 1966 feature more Fe and Mn, and less Ca. The varve preservation is also slightly worse. Although this is outside the scope of our investigation for this study, we interpret these changes to indicate that lake mixing was more intensive during these years. This may be due to a combination of climatic factors as well as changing vegetation. There has been reforestation around the lake since around 1950 (Wacnik et al., 2016), meaning the lake was more wind exposed in this time (1950-1965) compared to our study period. We have mentioned this change in the varve characteristics in the text (line 176) to clarify why we restricted the study period to 1966-2019, but we would prefer to exclude discussion of possible drivers of these changes as they are outside the scope of the present study.

Line 214 "High values of Ti denote the ice-covered period, when mainly fine lithogenic detrital material is deposited." Could you please briefly specify the process? Usually ice cover reduces sedimentation by reducing clastic material transport from the catchment and also protecting littoral sediments from waver activity and resuspension.

We will clarify that this is due to slow suspension settling of very fine (fine silt/clay-size) detritus. Additionally, biogenic production essentially stops under ice cover, which means the relative amount of clastic material settling is higher. This process is also well documented by comparisons of sediment trap monitoring and recent sediments in Bonk et al. (2015).

Line 252 Could you please specify how? This is less carefully explained compared to other three varve types.

We will expand on this VT-4 paragraph. VT-4 is generally similar to VT-2, with the most important difference being higher S throughout the varve year. We interpret this as an indication of strongly reducing conditions because oxidation would release sulfate into the overlying waters.

Line 316 ITRAX beam width 20mm?

We think this comment probably refers to line 416. For ITRAX it is 20-mm wide (across a core, parallel to strata). This was stated here as a contrast to the imaging technique we used. We will clarify that this was to highlight an advantage of the XRF imagining technique used in this study compared to conventional linescan core-scanning (ITRAX). The advantage is that we use a 2-mm wide window, which results in less mixing of layers (mixed pixels) in varves with boundaries that are tilted relative to the ITRAX window, or varves with otherwise complex geometry.

Figure 3 B: It would be nice to have years represented by each varve in addition to the information of the sediment depth. Can you add calendar years of each varve like you show them in fig 3A?

Yes, we will make this addition to the figure.

 **Technical corrections**

There is two discussion chapters in the manuscript: chapter 3 "results and discussion" and chapter 4 "discussion". This should be revised and structure clarified, by either having results and discussion at the same chapter or remove discussive parts from current "results and discussion" section and present them in chapter 4 in discussion. In my opinion both ways would work here.

Thank you for pointing out this error. Section 3 should be labeled "Results and interpretation"

**Authors' Response to RC2**

Thank you for your review and helpful comments. Our responses are in italics.

This paper reports a detailed, sophisticated and careful assessment of biochemical varves as an archive of past climate variability. In my view, the manuscript is a timely and important contribution to the varve palaeoclimate literature because it demonstrates convincingly that highresolution geochemistry (XRF) measurements and hyperspectral imaging can extract robust climatic signals at the seasonal scale from biochemical varves. I believe the research will be of great interest to the readership of Climate of thePast journal. The manuscript is well written and clear throughout and the statistical evaluation and analysis is convincing. I really commend the authors on this work. There are a handful of areas where clarification would be useful prior to publication but these are generally minor comments. Overall, the manuscript is, in my view, very suitable for publication.

**Comment #1:** Given the broad audience of the target journal (Climate of the Past), I suggest the authors incorporate a deeper overview of biochemical varves in the introduction. The use of clastic varves for similar purposes is mentioned on lines 38-40 so perhaps the next segment of text could say more on the key characteristics of biochemical varves.

We agree with reviewer's suggestion, and we have added background information on biochemical varves in the introduction.

**Comment #2:** It would be worth stipulating somewhere – and ideally early on – the basis for the calibration time window (1966 - 2019). Presumably this was guided by how far back in time meteorological data are available? Perhaps this could be explained at first use on Line 65.

The time window was mainly defined by the interval where we have the most confidence in the varve count (there is uncertainty in the count prior to 1965). Minimal (zero) chronological uncertainty is fundamental for the calibration of the varve-climate comparison. We have emphasized this in the introduction.

**Comment #3: T**he varve types are reported on Figures 2 and 3A prior to being explained in the text, which does not happen until Section 3.3. As a result, the coloured horizontal strips and legend rather lack context. I'm not sure what the best solution is; I agree it seems logical to present and interpret the long record first, as the authors have done, but the reader is left pondering the varve types. I suggest the authors give some thought to improving the sequencing here. Perhaps it would be most straightforward and sufficient to simply mention in the caption of the figures "see main text Section 3.3 for an explanation of the Varve Types"?

Yes, thank you, your suggested addition to the caption seems suitable and helpful.

**Comment #4:** The decadal variability in the dominant varve type is striking and intriguing. For example, the authors draw attention to the notable shift away from VT3 after the late-1980s (e.g. Line 238). I would welcome some commentary from the authors on the following aspects: what are the potential implications of decadal variability on centennial or millennial-scale varve-based climate reconstructions? Similarly, how different would the regressions and calibrations be if, say, the instrumental data used in the calibration only extended back to the late 1980s, thereby missing the period when VT3 is the dominant signal? How different would the GAM outputs be if only the VT-3 data were used?

We interpret the decadal variability in VT to be significantly driven by climate, though other factors can of course play a role. Therefore, we think our data shows a strong potential for varve-based climate reconstructions to capture decadal variability, though longer datasets would be required to conclude this with confidence.

To some extent, the split-period validation touches upon the second part of this comment. We found that the GAM outputs for the temperature model were generally similar for the periods 1966-1992 and 1993-2019. The wind model, however, changed substantially when fit to data

from 1966-1991 versus data from 1992 and 1995-2019 (1993-1994 missing data). We will do further testing on VT-3 only data to see the effect on the GAM output and report this in the final revision report.

[Figure]

*Figure: Comparison of the full GAM outputs with models trained on Vty-3 only data and Vty-1,-2,-3 data. Top: MAMJJA temperature model, bottom: Mar-Dec wind days model.*

Based on the reviewers comment, we have conducted an analysis of GAM outputs using data trained on the Vty-3 years or the non-Vty-3 years. We find that the models based on smaller subsets reasonably agree with the full model in terms of decadal-scale trends, suggesting that the shift in varve type around 1987 is not a dominant control on the results of the GAM outputs.

However, it is worth noting that the Vty-3 model shows a much stronger effect from Ti compared to TC and vice versa for the non-Vty-3 model. This leads to dampened annual variability in the non-Vty-3 model due to less year-to-year variability in TC compared to Ti.

In our opinion, the reviewer's questions about how different subsets of instrumental data affect the calibrations and resulting models can be assessed by the split-period validation briefly discussed in the manuscript. Rather than include this new Vty-3 data-splitting exercise in the manuscript in addition to the existing data splitting approach we used (simply dividing the dataset into equal halves), we prefer to focus on the original split-period analysis. We have made the following modifications: we added a supplemental figure that shows the GAM outputs of the split-period models (similar to the figure above), added text to the methods to clarify what was done for the split-period validation, and expanded our discussion of the stability (or lack of stability) of climate-proxy relationships in section 3.5.

**Comment #5:** The data and statistical output are convincing but the presentational format of the correlation coefficients is a bit odd and inconsistent. Figure 6 (monthly correlations) works nicely in terms of aesthetics and reporting a great deal of data in a simple and effective way. I was left wondering about seasonal correlations, which I then find in Table 1. The table is fine but it is more difficult to trace the positive and negative correlations across sedimentary variables and seasons – not least because significant positive and significant negative correlations are in bold font. Did the authors have a specific reason for using one figure and one table? I suggest they consider plotting both as figures using the same formatting style.

We have added a plot of seasonal correlations to Fig. 6. We would like to also include Table 1 so that people can easily read the r values.

**Technical comments:**

**Figure 1A:** It may just be my screen but the light and dark green colours were difficult to distinguish.

We have enhanced the contrast of the map.

**Comment:** Superscript formatting is missing in a handful of instances, for example MAR units on Figure 2 and 137Cs in the figure caption.

Thank you for pointing out these formatting errors, they have been corrected.

**Line 416:** Is the XRF beam 20-mm wide?

For ITRAX it is 20-mm wide (across a core, parallel to strata). This was stated here as a contrast to the imaging technique we used. The advantage is that we use a 2-mm wide window, which results in less mixing of layers (mixed pixels) in varves with boundaries that are tilted relative to the ITRAX window, or varves with otherwise complex geometry. We have clarified this in the text (line 433).

**Authors' Response to EC1**

Dear Authors,

I have read the reviewers comments, and both seems quite positive about your manuscript. I also read your preprint, and I also found it well written, structured and presenting a very promising new way to analyze varved sediments.

Thank you for your comments, we will implement these suggestions in the revised manuscript. Our responses are provided in red text.

You will find below a few minor comments or suggestions to improve your manuscript.

Lines 69-71: this sentence belongs to a conclusion not to an introduction.

We have removed this sentence from the introduction.

Lines 99-100: can you provide more information about the dry bulk density sampling technique for each varve? Your thick varves are 6 mm thick, but this is still quite thin for discrete sampling.

Material was sampled carefully from within a single varve, packed into a 3 cm$^3$ syringe and weighed. The material was then dried and weighed once again to determine dry bulk density. We have added this information to line 99.

Lines 128-129: Can you provide this information in Supplement?

Yes, we can include a table of corrected and uncorrected data in the supplement.

Line 134: this link to data is currently not working, but I was able to access them using this link: https://boris.unibe.ch/156383/. Please make sure to provide a working link in the revised version.

The DOI link is working for us as of this writing. Please contact me if it is still not working for you.

Lines 92-93: Can you better explain what you mean by the onset of varve precipitation, and indicate that point in a Ca profile (for instance in Figure S1).

We have added text here to clarify. Varve boundaries were assigned using the Ca data images (maps) where the varve boundary marked the onset of Ca-rich lamina. This was done with support from thin section images and XRF data to ensure the correct placement of the boundary. We now show an example with the varve boundaries on the Ca data in Fig. S2.

Line 139 and following: I understand that you can have the entire liberty to choose your own abbreviation for Varve Types (VT). However, VT is often used for Varve Thickness, and this may lead to some confusion. Maybe could you change VT for VTy or something else, but this is really a suggestion and you may decide to keep VT.

This is a good point since VT is widely used as varve thickness. We have changed VT to Vty throughout.

Line 143: You write here that "Data were detrended, log-transformed and normalized prior to classification". Were data detrended, log-transformed and normalized prior to the perform the alignment of the two sets of data (µXRF and HSI) or did you use raw data for that alignment? I think some clarification is required here. By the way, this is very interesting.

Detrending, log-transformation, and normalization were only done for the varve type classification analysis. The data were normalized before the alignment step, simply for the purpose of being able to plot the data on the same scale. Correlations and GAMs were done using raw data. We have clarified this in the methods text.

Lines 157-159: maybe a reference that covers the GAMs?

Yes, Wood, 2017 should be added here.

Line 174: You write, "The 137Cs activities in these two varves are indistinguishable within the measurement uncertainty". Maybe you should write, "The 137Cs activities in these two varves are indistinguishable from each other within the measurement uncertainty.

We agree with the modification.

Line 283: Why do you add this argument, why mentioning carbon burial? You mean carbon trapped by carbonate or organic carbon?

Here, we had in mind mainly organic carbon, because previous studies have shown it is better preserved under stronger thermal stratification. We included this argument as it could be a secondary mechanism driving the positive correlations seen between temperature and TC.

Figure 3: on the legend of the horizontal axis, µXRF (counts per second), add µ. Also, there are values less than 1 in the µXRF data, meaning that these counts were normalized somehow. Can you be more explicit about that?

Yes, we will add µ. The reason for fractional count data is that for some elements there was less than 1 count per second per pixel (or measurement spot). The only normalization applied was division by counting time. Each data point plotted represents the average counts per second of each pixel within a 60 x 2000 µm area. Summing the counts across pixels before normalizing by measurement time might result in more intuitive numeric values, but would not change the results.

Figure S2. What are the dots pointing to? Please specify in the caption.

We will add to the caption that dots mark varve boundaries.

So the next step for you is to reply to the comments of the reviewers and mine using the Copernicus system, maybe not in detail, but enough to allow me to get to the step of the process, i.e., authorizing you to submit a revised version.

I'm looking forward to your replies.

Best regards,

Pierre Francus

**References**

Bonk, A., Tylmann, W., Amann, B., Enters, D. and Grosjean, M.: Modern limnology and varve-formation processes in lake Żabińskie, northeastern Poland: Comprehensive process studies as a key to understand the sediment record, J. Limnol., 74(2), 358–370, doi:10.4081/jlimnol.2014.1117, 2015.

Wacnik, A., Tylmann, W., Bonk, A., Goslar, T., Enters, D., Meyer-Jacob, C. and Grosjean, M.: Determining the responses of vegetation to natural processes and human impacts in north-eastern Poland during the last millennium: combined pollen, geochemical and historical data, Veg. Hist. Archaeobot., 25(5), 479–498, doi:10.1007/s00334-016-0565-z, 2016.

Żarczyński, M., Tylmann, W. and Goslar, T.: Multiple varve chronologies for the last 2000 years from the sediments of Lake Żabińskie (northeastern Poland) – Comparison of strategies for varve counting and uncertainty estimations, Quat. Geochronol., 47(January), 107–119, doi:10.1016/j.quageo.2018.06.001, 2018.

---

## Author Response (AR2)

Dear Pierre,

Thank you for pointing out this misunderstanding regarding the ITRAX detector. We have removed the sentences discussing the width of the measurement area. We have also corrected the typo you mentioned in the caption of Figure S12. Thank you for a smooth review process and your kind words about our study.

On behalf of the authors,

Paul Zander